# PERSONALIZED FEDERATED LEARNING WITH FIRST ORDER MODEL OPTIMIZATION

**Michael Zhang**[*]
Stanford University
mzhang@cs.stanford.edu

**Karan Sapra**
NVIDIA
ksapra@nvidia.com

**Sanja Fidler**
NVIDIA
sfidler@nvidia.com

**Serena Yeung**
Stanford University
syyeung@stanford.edu

**Jose M. Alvarez**
NVIDIA
josea@nvidia.com

## ABSTRACT

While federated learning traditionally aims to train a single global model across decentralized local datasets, one model may not always be ideal for all participating clients. Here we propose an alternative, where each client only federates with other relevant clients to obtain a stronger model per client-specific objectives. To achieve this personalization, rather than computing a single model average with constant weights for the entire federation as in traditional FL, we efficiently calculate optimal weighted model combinations for each client, based on figuring out how much a client can benefit from another's model. We do not assume knowledge of any underlying data distributions or client similarities, and allow each client to optimize for arbitrary target distributions of interest, enabling greater flexibility for personalization. We evaluate and characterize our method on a variety of federated settings, datasets, and degrees of local data heterogeneity. Our method outperforms existing alternatives, while also enabling new features for personalized FL such as transfer outside of local data distributions.

## 1 INTRODUCTION

Federated learning (FL) has shown great promise in recent years for training a single global model over decentralized data. While seminally motivated by effective inference on a general test set similar in distribution to the decentralized data in aggregate (McMahan et al., 2016; Bonawitz et al., 2019), here we focus on federated learning from a *client-centric* or *personalized* perspective. We aim to enable stronger performance on personalized target distributions for each participating client. Such settings can be motivated by *cross-silo* FL, where clients are autonomous data vendors (e.g. hospitals managing patient data, or corporations carrying customer information) that wish to collaborate without sharing private data (Kairouz et al., 2019). Instead of merely being a source of data and model training for the global server, clients can then take on a more active role: their federated participation may be contingent on satisfying client-specific target tasks and distributions. A strong FL framework in practice would then flexibly accommodate these objectives, allowing clients to optimize for arbitrary distributions simultaneously in a single federation.

In this setting, FL's realistic lack of an independent and identically distributed (IID) data assumption across clients may be both a burden and a blessing. Learning a single global model across non-IID data batches can pose challenges such as non-guaranteed convergence and model parameter divergence (Hsieh et al., 2019; Zhao et al., 2018; Li et al., 2020). Furthermore, trying to fine-tune these global models may result in poor adaptation to local client test sets (Jiang et al., 2019). However, the non-IID nature of each client's local data can also provide useful signal for distinguishing their underlying local data distributions, without sharing any data. We leverage this signal to propose a new framework for personalized FL. Instead of giving all clients the same global model average weighted by local training size as in prior work (McMahan et al., 2016), for each client we compute

---

[*]Corresponding author; work done while interning at NVIDIA

a weighted combination of the available models to best align with that client's interests, modeled by evaluation on a personalized target test distribution.

Key here is that after each federating round, we maintain the client-uploaded parameters individually, allowing clients in the next round to download these copies independently of each other. Each federated update is then a two-step process: given a local objective, clients (1) evaluate how well their received models perform on their target task and (2) use these respective performances to weight each model's parameters in a personalized update. We show that this intuitive process can be thought of as a particularly coarse version of popular iterative optimization algorithms such as SGD, where instead of directly accessing other clients' data points and iteratively training our model with the granularity of gradient decent, we limit ourselves to working with their uploaded models. We hence propose an efficient method to calculate these optimal combinations for each client, calling it `FedFomo`, as (1) each client's federated update is calculated with a simple first-order model optimization approximating a personalized gradient step, and (2) it draws inspiration from the "fear of missing out", every client no longer necessarily factoring in contributions from all active clients during each federation round. In other words, curiosity *can* kill the cat. Each model's personalized performance can be saved however by restricting unhelpful models from each federated update.

We evaluate our method on federated image classification and show that it outperforms other methods in various non-IID scenarios. Furthermore, we show that because we compute federated updates directly with respect to client-specified local objectives, our framework can also optimize for *out-of-distribution* performance, where client's target distributions are different from their local training ones. In contrast, prior work that personalized based on similarity to a client's own model parameters (Mansour et al., 2020; Sattler et al., 2020) restricts this optimization to the local data distribution. We thus enable new features in personalized FL, and empirically demonstrate up to $70\%$ improvement in some settings, with larger gains as the number of clients or level of non-IIDness increases.

**Our contributions**

1. We propose a flexible federated learning framework that allows clients to personalize to specific target data distributions irrespective of their available local training data.

2. Within this framework, we introduce a method to efficiently calculate the optimal weighted combination of uploaded models as a personalized federated update

3. Our method strongly outperforms other methods in non-IID federated learning settings.

## 2 Related Work

**Federated Learning with Non-IID Data**  While fine-tuning a global model on a client's local data is a natural strategy to personalize (Mansour et al., 2020; Wang et al., 2019), prior work has shown that non-IID decentralized data can introduce challenges such as parameter divergence (Zhao et al., 2018), data distribution biases (Hsieh et al., 2019), and unguaranteed convergence Li et al. (2020). Several recent methods then try to improve the robustness of global models under heavily non-IID datasets. `FedProx` (Li et al., 2020) adds a proximal term to the local training objective to keep updated parameter close to the original downloaded model. This serves to reduce potential weight divergence defined in Zhao et al. (2018), who instead allow clients to share small subsets of their data among each other. This effectively makes each client's local training set closer in distribution to the global test set. More recently, Hsu et al. (2019) propose to add momentum to the global model update in `FedAvgM` to reduce the possibly harmful oscillations associated with averaging local models after several rounds of stochastic gradient descent for non-identically distributed data.

While these advances may make a global model more robust across non-IID local data, they do not directly address local-level data distribution performance relevant to individual clients. Jiang et al. (2019) argue this latter task may be more important in non-IID FL settings, as local training data differences may suggest that only a subset of all potential features are relevant to each client. Their target distributions may be fairly different from the global aggregate in highly personalized scenarios, with the resulting dataset shift difficult to handle with a single model.

**Personalized Federated Learning**  Given the challenges above, other approaches train multiple models or personalizing components to tackle multiple target distributions. Smith et al. (2017) propose multi-task learning for FL with `MOCHA`, a distributed MTL framework that frames clients as tasks and learns one model per client. Mixture methods (Deng et al., 2020; Hanzely & Richtárik,

2020; Mansour et al., 2020) compute personalized combinations of model parameters from training both local models and the global model, while Peterson et al. (2019) ensure that this is done with local privacy guarantees. Liang et al. (2020) apply this mixing across network layers, with lower layers acting as local encoders that map a client's observed data to input for a globally shared classifier. Rather than only mix with a shared global model, our work allows for greater control and distinct mixing parameters with multiple local models. Fallah et al. (2020) instead optimize the global model for fast personalization through meta-learning, while T Dinh et al. (2020) train global and local models under regularization with Moreau envelopes. Alternatively, Clustered FL (Sattler et al., 2020; Ghosh et al., 2020; Briggs et al., 2020; Mansour et al., 2020) assumes that inherent partitions or data distributions exist behind clients' local data, and aim to cluster these partitions to federate within each cluster. Our work does not restrict which models are computed together, allowing clients to download suitable models independently. We also compute client-specific weighted averages for greater personalization. Finally, unlike prior work, we allow clients to receive personalized updates for target distributions different from their local training data.

## 3 FEDERATED FIRST ORDER MODEL OPTIMIZATION

We now present `FedFomo`, a personalized FL framework to efficiently compute client-optimizing federated updates. We adopt the general structure of most FL methods, where we iteratively cycle between downloading model parameters from server to client, training the models locally on each client's data, and sending back the updated models for future rounds. However, as we do not compute a single global model, each federated download introduces two new steps: (1) figuring out which models to send to which clients, and (2) computing their personalized weighted combinations. We define our problem and describe how we accomplish (1) and (2) in the following sections.

**Problem Definition and Notation**   Our work most naturally applies to heterogeneous federated settings where participating clients are critically not restricted to single local training or target test distribution, and apriori we do not know anything about these distributions. To model this, let $\mathbb{C}$ be a population with $|\mathbb{C}| = K$ total clients, where each client $c_i \in \mathbb{C}$ carries local data $\boldsymbol{D}_i$ sampled from some distribution $\mathcal{D}$ and local model parameters $\theta_i^{\ell(t)}$ during any round $t$. Each $c_i$ also maintains some personalized objective or task $\mathcal{T}_i$ motivating their participation in the federation. We focus on supervised classification as a universal task setting. Each client and task are then associated with a test dataset $\boldsymbol{D}_i^{\text{test}} \sim \mathcal{D}^*$. We define each $\mathcal{T}_i := \min \mathcal{L}(\theta_i^{\ell(t)}; \boldsymbol{D}_i^{\text{test}})$, where $\mathcal{L}(\theta; \boldsymbol{D}) : \Theta \mapsto \mathbb{R}$ is the loss function associated with dataset $\boldsymbol{D}$, and $\Theta$ denotes the space of models possible with our presumed network architecture. We assume no knowledge regarding clients and their data distributions, nor that test and local data belong to the same distribution. We aim to obtain the optimal set of model parameters $\{\theta_1^*, \ldots, \theta_K^*\} = \arg\min \sum_{i \in [K]} \mathcal{L}_{\mathcal{T}_i}(\theta_i)$.

### 3.1 COMPUTING FEDERATED UPDATES WITH FOMO

Unlike previous work in federated learning, `FedFomo` learns optimal combinations of the available server models for each participating client. To do so, we leverage information from clients in two different ways. First, we aim to directly optimize for each client's target objective. We assume that clients can distinguish between good and bad models on their target tasks, through the use of a labeled validation data split $\boldsymbol{D}_i^{\text{val}} \subset \boldsymbol{D}_i$ in the client's local data. $\boldsymbol{D}_i^{\text{val}}$ should be similar in distribution to the target test dataset $\boldsymbol{D}_i^{\text{test}}$. The client can then evaluate any arbitrary model $\theta_j$ on this validation set, and quantify the performance through the computed loss, denoted by $\mathcal{L}_i(\theta_j)$. Second, we directly leverage the potential heterogeneity among client models. Zhao et al. (2018) explore this phenomenon as a failure mode for traditional single model FL, where they show that diverging model weights come directly from local data heterogeneity. However, instead of combining these parameters into a single global model, we maintain the uploaded models individually as a means to preserve a model's potential contribution to another client. Critically, these two ideas together not only allow us to compute more personal model updates within non-IID local data distributions, but also enable clients to optimize for data distributions different from their own local data's.

**Federated learning as an iterative local model update**   The central premise of our work stems from viewing each federated model download−and subsequent changing of local model parameters−as an optimization step towards some objective. In traditional FL, this objective involves performing well on the global population distribution, similar in representation to the union of all local datasets. Assuming $N$ federating clients, we compute each global model $\theta^G$ at time $t$ as:

$\theta^{G(t)} = \sum_{n=1}^{N} w_n \cdot \theta_n^{\ell(t)}$, where $w_n = |\boldsymbol{D}_n^{\text{train}}| / \sum_{j=1}^{N} |\boldsymbol{D}_j^{\text{train}}|$. If client $c_i$ downloads this model, we can view this change to their local model as an update: $\theta_i^{\ell(t+1)} \leftarrow \theta_i^{\ell(t)} + \sum_{n=1}^{N} w_n \cdot (\theta_n^{\ell(t)} - \theta_i^{\ell(t)})$ since $\sum_n w_n = 1$. This then updates a client's current local model parameters in directions specified by the weights $\boldsymbol{w}$ and models $\{\theta_n\}$ in the federation. A natural choice to optimize for the global target distribution sets $w_n$ as above and in McMahan et al. (2017), e.g. as an unbiased estimate of global model parameters. However, in our personalized scenario, we are more interested in computing the update uniquely with respect to each client's target task. We then wish to find the optimal weights $\boldsymbol{w} = \langle w_1, \ldots, w_N \rangle$ that optimize for the client's objective, minimizing $\mathcal{L}_i(\theta_i^\ell)$.

**Efficient personalization with FedFomo** Intuitively, we wish to find models $\{\theta_m^{\ell(t)} : m \in [N] \setminus i\}$ such that moving towards their parameters leads to better performance on our target distribution, and accordingly weight these $\theta$ higher in a model average. If a client carries a satisfying number of local data points associated with their target objective $\mathcal{L}_i$, then they could obtain a reasonable model through local training alone, e.g. directly updating their model parameters through SGD:

$$\theta_i^{\ell(t+1)} \leftarrow \theta_i^{\ell(t)} - \alpha \nabla_\theta \mathcal{L}_i(\theta_i^{\ell(t)}) \tag{1}$$

However, without this data, clients are more motivated to federate. In doing so they obtain useful updates, albeit in the more restricted form of fixed model parameters $\{\theta_n : n \in N\}$. Then for personalized or non-IID target distributions, we can iteratively solve for the optimal combination of client models $\boldsymbol{w}^* = \arg\min \mathcal{L}_i(\theta)$ by computing:

$$\theta_i^{\ell(t+1)} \leftarrow \theta_i^{\ell(t)} - \alpha \boldsymbol{1}^\top \nabla_{\boldsymbol{w}} \mathcal{L}_i(\theta_i^{\ell(t)}) \tag{2}$$

where $\boldsymbol{1}$ is a size-$N$ vector of ones. Unfortunately, as the larger federated learning algorithm is already an iterative process with many rounds of communication, computing $\boldsymbol{w}^*$ through Eq. 2 may be cumbersome. Worse, if the model averages are only computed server-side as in traditional FL, Eq. 2 becomes prohibitively expensive in communication rounds (McMahan et al., 2017).

Following this line of reasoning however, we thus derive an approximation of $\boldsymbol{w}^*$ for any client: Given previous local model parameters $\theta_i^{\ell(t-1)}$, set of fellow federating models available to download $\{\theta_n^{\ell(t)}\}$ and local client objective captured by $\mathcal{L}_i$, we propose weights of the form:

$$w_n = \frac{\mathcal{L}_i(\theta_i^{\ell(t-1)}) - \mathcal{L}_i(\theta_n^{\ell(t)})}{\|\theta_n^{\ell(t)} - \theta_i^{\ell(t-1)}\|} \tag{3}$$

where the resulting federated update $\theta_i^{\ell(t)} \leftarrow \theta_i^{\ell(t-1)} + \sum_{n \in [N]} w_n(\theta_n^{\ell(t)} - \theta_i^{\ell(t-1)})$ directly optimizes for client $c_i$'s objective up to a first-order approximation of the optimal $\boldsymbol{w}^*$. We default to the original parameters $\theta_i^{\ell(t-1)}$ if $w_n < 0$ above, i.e. $w_n = \max(w_n, 0)$, and among positive $w_n$ normalize to get final weights $w_n^* = \frac{\max(w_n, 0)}{\sum_n \max(w_n, 0)}$ to maintain $w^* \in [0, 1]$ and $\sum_{n=1}^{N} w_n^* \in \{0, 1\}$.

We derive Eq. 3 as a first-order approximation of $\mathbf{w}^*$ in Appendix A.1. Here we note that our formulation captures the intuition of federating with client models that perform better than our own model, e.g. have a smaller loss on $\mathcal{L}_i$. Moreso, we weigh models more heavily as this positive loss delta increases, or the distance between our current parameters and theirs decreases, in essence most heavily weighing the models that most efficiently improve our performance. We use local parameters at $t$-1 to directly compute how much we should factor in current parameters $\theta_i^{\ell(t)}$, which also helps prevent overfitting as $\mathcal{L}_i(\theta_i^{\ell(t-1)}) - \mathcal{L}_i(\theta_i^{\ell(t)}) < 0$ causes "early-stopping" at $\theta_i^{\ell(t-1)}$.

**Communication and bandwidth overhead** Because the server can send multiple requested models in one download to any client, we still maintain one round of communication for model downloads and one round for uploads in between $E$ local training epochs. Furthermore, because $\boldsymbol{w}$ in Eq. 3 is simple to calculate, the actual model update can also happen client-side, keeping the total number of communications with $T$ total training epochs at $\lfloor \frac{2T}{E} \rfloor$, as in `FedAvg`.

However `FedFomo` also needs to consider the additional bandwidth from downloading multiple models. While quantization and distillation (Chen et al., 2017; Hinton et al., 2015; Xu et al., 2018) can alleviate this, we also avoid worst case $N^2$ overhead with respect to the number of active clients

$N$ by restricting the number of models downloaded $M$. Whether we can achieve good personalization here involves figuring out which models benefit which clients, and our goal is then to send as many helpful models as possible given limited bandwidth.

To do so, we invoke a sampling scheme where the likelihood of sending model $\theta_j$ to client $c_i$ relies on how well $\theta_j$ performed regarding client $c_i$'s target objective in previous rounds. Accordingly, we maintain an affinity matrix $\boldsymbol{P}$ composed of vectors $\boldsymbol{p}_i = \langle p_{i,1}, \ldots, p_{i,K} \rangle$, where $p_{i,j}$ measures the likelihood of sending $\theta_j$ to client $c_i$, and at each round send the available uploaded models corresponding to the top $M$ values according to each participating client's $\boldsymbol{p}$. Initially we set $\boldsymbol{P} = \mathrm{diag}(1, \ldots, 1)$, i.e. each model has an equal chance of being downloaded. Then during each federated update, we update $\boldsymbol{p} \leftarrow \boldsymbol{p} + \boldsymbol{w}$ from Eq. 3, where $\boldsymbol{w}$ can now be negative. If $N \ll K$, we may benefit from additional exploration, and employ an $\varepsilon$-greedy sampling strategy where instead of picking strictly in order of $\boldsymbol{p}$, we have $\varepsilon$ chance to send a random model to the client. We investigate the robustness of `FedFomo` to these parameters through ablations of $\varepsilon$ and $M$ in the next section.

## 4 EXPERIMENTS

**Experimental Setup**   We consider two different scenarios for simulating non-identical data distributions across federating clients. First we evaluate with the *pathological* non-IID setup in McMahan et al. (2016), where each client is randomly assigned 2 classes among 10 total classes. We also use a *latent distribution* non-IID setup, where we first partition our datasets based on feature and semantic similarity, and then sample from them to setup different local client data distributions. We use number of distributions $\in \{2, 3, 4, 5, 10\}$ and report the average Earth Mover's Distance (EMD) between local client data and the total dataset across all clients to quantify non-IIDness. We evenly allocate clients among distributions and include further details in Appendix A.5. We evaluate under both setups with two FL scenarios: 15 and 100 clients with $100\%$ and $10\%$ participation respectively, reporting final accuracy after training with $E = 5$ local epochs per round for 20 communication rounds in the former and 100 rounds in the latter. Based on prior work (McMahan et al., 2016; Liang et al., 2020), we compare methods with the MNIST (LeCun et al., 1998), CIFAR-10 (Krizhevsky et al., 2009), and CIFAR-100 datasets. We use the same CNN architecture as in McMahan et al. (2016).

**Federated Learning Baselines**   We compare `FedFomo` against methods broadly falling under two categories: they (1) propose modifications to train a single global model more robust to non-IID local datasets, or (2) aim to train more than one model or model component to personalize performance directly to client test sets. For (1), we consider `FedAvg`, `FedProx`, and the 5% data-sharing strategy with `FedAvg`, while in (2) we compare our method to `MOCHA`, `LG-FedAvg`, `Per-FedAvg`, `pFedMe`, Clustered Federated Learning (`CFL`), and a local training baseline. All accuracies are reported with mean and standard deviation over three runs, with local training epochs $E = 5$, the same number of communication rounds (20 for 15 clients, $100\%$ participation; 100 for 100 clients, $10\%$ participation) and learning rate 0.01 for MNIST, 0.1 for CIFAR-10). We implemented all results[1].

**Pathological Non-IID**   We follow precedent and report accuracy after assigning two classes out of the ten to each client for the pathological setting in Table 1. Across datasets and client setups, our proposed `FedFomo` strongly outperforms alternative methods in settings with larger number clients, and achieves competitive accuracy in the 15 client scenario. In the larger 100 client scenario, each individual client participates less frequently but also carries less local training data. Such settings motivate a higher demand for efficient federated updates, as there are less training rounds for each client overall. Meanwhile, methods that try to train a single robust model perform with mixed success over the `FedAvg` baseline, and notably do not perform better than local training alone. Despite the competitive performance, we note that this pathological setting is not the most natural scenario to apply `FedFomo`. In particular when there are less clients, each client's target distribution carries only 2 random classes, there is no guarantee that any two clients share the same objective such that they can clearly benefit each other. With more clients however, we can also expect higher frequencies of target distribution overlap, and accordingly find that we outperform all other methods.

**Latent Distribution Non-IID**   We next report how each FL method performs in the latent distribution setting in Table 2, with additional results in Fig. 1. Here we study the relative performance of

---

[1]`LG-FedAvg` and `MOCHA` were implemented with code from github.com/pliang279/LG-FedAvg. `pFedMe` and `Per-FedAvg` were implemented with code from github.com/CharlieDinh/pFedMe. `CFL` was implemend with code from github.com/felisat/clustered-federated-learning

|  | MNIST | | CIFAR-10 | |
| --- | --- | --- | --- | --- |
|  | 15 clients | 100 clients | 15 clients | 100 clients |
| Local Training | $99.62 \pm 0.21$ | $94.25 \pm 4.28$ | $92.73 \pm 0.87$ | $85.42 \pm 4.06$ |
| FedAvg (McMahan et al., 2016) | $91.97 \pm 2.17$ | $78.83 \pm 9.68$ | $57.12 \pm 1.14$ | $53.08 \pm 7.40$ |
| FedAvg + Data (Zhao et al., 2018) | $91.99 \pm 2.14$ | $78.85 \pm 9.66$ | $58.5 \pm 1.67$ | $56.62 \pm 8.92$ |
| FedProx (Li et al., 2020) | $90.94 \pm 1.85$ | $79.06 \pm 10.88$ | $54.60 \pm 3.26$ | $52.92 \pm 5.56$ |
| LG-FedAvg (Liang et al., 2020) | $\mathbf{99.79 \pm 0.07}$ | $84.17 \pm 1.92$ | $92.36 \pm 1.00$ | $84.17 \pm 4.45$ |
| MOCHA (Smith et al., 2017) | $94.74 \pm 2.27$ | $84.58 \pm 5.80$ | $\mathbf{93.85 \pm 2.04}$ | $76.09 \pm 8.49$ |
| Clustered FL (CFL) (Sattler et al., 2020) | $95.00 \pm 3.61$ | $92.26 \pm 3.91$ | $85.07 \pm 8.16$ | $77.75 \pm 1.78$ |
| Per-FedAvg (Fallah et al., 2020) | $92.39 \pm 4.72$ | $85.32 \pm 12.93$ | $81.96 \pm 8.12$ | $72.40 \pm 4.06$ |
| pFedMe (T Dinh et al., 2020) | $97.70 \pm 1.26$ | $88.40 \pm 10.86$ | $83.85 \pm 5.11$ | $71.75 \pm 6.78$ |
| Ours (5 clients downloaded) | $99.62 \pm 2.91$ | $\mathbf{98.81 \pm 1.26}$ | $93.01 \pm 0.96$ | $92.10 \pm 5.20$ |
| Ours (10 clients downloaded) | $99.63 \pm 0.07$ | $98.71 \pm 2.86$ | $92.73 \pm 0.96$ | $\mathbf{92.67 \pm 4.21}$ |

Table 1: Personalized FL accuracy with pathological non-IID splits. Best results in bold. `FedFomo` outperforms or is competitive with prior work across settings, especially with larger populations.

`FedFomo` across various levels of statistical heterogeneity, and again show that our method strongly outperforms others in highly non-IID settings. The performance gap widens as local datasets become more non-IID, where global FL methods may suffer more from combining increasingly divergent weights while also experiencing high target data distribution shift (quantified with higher EMD) due to local data heterogeneity. Sharing a small amount of data among clients uniformly helps, as does actively trying to reduce this divergence through `FedProx`, but higher performance most convincingly come from methods that do not rely on a single model. The opposite trend occurs with local training, as more distributions using the same 10 or 100 classes leads to smaller within-distribution variance. Critically, `FedFomo` is competitive with local training in the most extreme non-IID case while strongly outperforming `FedAvg`, and outperforms both in moderately non-IID settings (EMD $\in [1, 2]$), suggesting that we can selectively leverage model updates that best fit client objectives to justify federating. When data is more IID, any individual client model may benefit another, and it becomes harder for a selective update to beat a general model average. `FedFomo` also outperforms personalizing-component and multi-model approaches (`MOCHA` and `LG-FedAvg`), where regarding data heterogeneity we see similar but weaker and more stochastic trends in performance.

|  | CIFAR-10 Number of Latent Distributions (EMD) | | | | |
| --- | --- | --- | --- | --- | --- |
| Method | 2 (1.05) | 3 (1.41) | 4 (1.28) | 5 (2.80) | 10 (2.70) |
| Local Training | $60.03 \pm 9.22$ | $66.61 \pm 9.90$ | $69.12 \pm 12.07$ | $76.52 \pm 11.46$ | $92.64 \pm 7.32$ |
| FedAvg | $38.92 \pm 11.88$ | $21.56 \pm 9.14$ | $22.34 \pm 12.36$ | $32.13 \pm 1.95$ | $10.10 \pm 3.65$ |
| FedAvg + Data | $53.43 \pm 2.89$ | $33.87 \pm 2.53$ | $65.73 \pm 1.07$ | $63.32 \pm 0.49$ | $41.61 \pm 0.92$ |
| FedProx | $66.42 \pm 1.79$ | $31.38 \pm 2.54$ | $50.61 \pm 1.53$ | $48.20 \pm 0.14$ | $13.41 \pm 3.39$ |
| LG-FedAvg | $70.87 \pm 1.12$ | $74.16 \pm 2.37$ | $67.25 \pm 1.97$ | $63.64 \pm 2.52$ | $94.42 \pm 1.25$ |
| MOCHA | $\mathbf{83.79 \pm 1.54}$ | $73.68 \pm 2.80$ | $71.23 \pm 4.08$ | $69.02 \pm 2.93$ | $94.28 \pm 0.81$ |
| CFL | $72.58 \pm 10.30$ | $75.69 \pm 1.11$ | $78.31 \pm 12.90$ | $70.04 \pm 13.56$ | $85.22 \pm 6.70$ |
| Per-FedAvg | $63.85 \pm 5.11$ | $69.70 \pm 7.27$ | $72.60 \pm 9.28$ | $76.61 \pm 6.65$ | $93.97 \pm 2.34$ |
| pFedMe | $49.87 \pm 3.16$ | $66.95 \pm 10.65$ | $69.00 \pm 4.97$ | $78.66 \pm 3.72$ | $94.57 \pm 1.95$ |
| Ours (n=5) | $77.823 \pm 2.24$ | $82.38 \pm 0.66$ | $\mathbf{84.45 \pm 0.21}$ | $85.050 \pm 0.13$ | $95.55 \pm 0.26$ |
| Ours (n=10) | $79.59 \pm 0.34$ | $\mathbf{83.66 \pm 0.72}$ | $84.35 \pm 0.38$ | $\mathbf{85.534 \pm 0.53}$ | $\mathbf{95.55 \pm 0.06}$ |

Table 2: In-distribution federated accuracy with 15 clients, 100% participation, across heterogeneity levels (measured by EMD). `FedFomo` performs better than or competitively with existing methods.

**Personalized model weighting** We next investigate `FedFomo`'s personalization by learning optimal client to client weights overtime, visualizing $\boldsymbol{P}$ during training in Fig. 2. We depict clients with the same local data distributions next to each other (e.g. clients $0, 1, 2$ belong to distribution $0$). Given the initial diagonal $\boldsymbol{P}$ depicting equal weighting for all other clients, we hope `FedFomo` increases the weights of clients that belong to the same distribution, discovering the underlying partitions without knowledge of client datasets. In Fig 2a we show this for the 15 client 5 non-IID latent distribution setting on CIFAR-10 with 5 clients downloaded and $\varepsilon = 0.3$ (lighter = higher weight). These default parameters adjust well to settings with more total clients (Fig 2b), and when we change the number of latent distributions (and IID-ness) in the federation (Fig 2c).

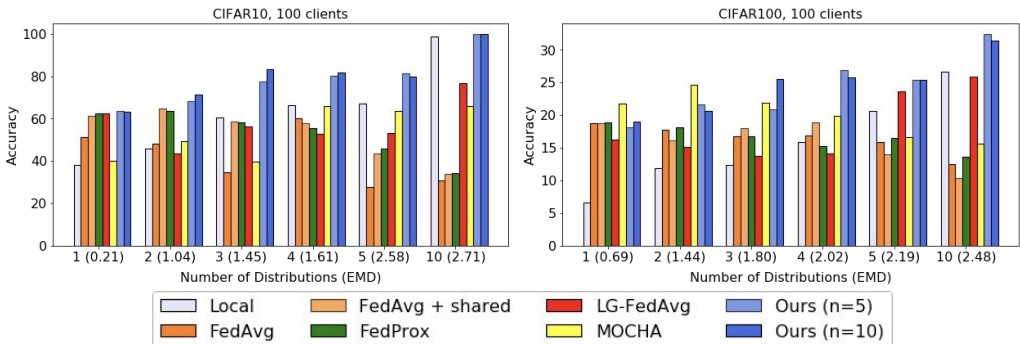

Figure 1: Classification accuracy of FL frameworks with 100 clients over latent distributions.

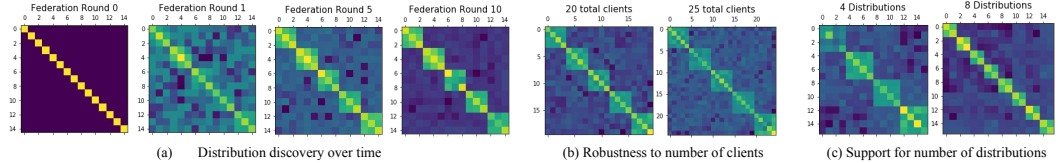

Figure 2: FedFomo client-to-client weights over time and across different FL settings. We reliably upweight clients with the same training and target distributions.

**Exploration with $\varepsilon$ and number of models downloaded $M$** To further understand FedFomo's behavior and convergence in non-IID personalized settings with respect to limited download bandwidth capability, we conduct an ablation over $\varepsilon$ and $M$, reporting results on the 15 client CIFAR-10 5-distribution setting in Fig. 3 over 100 training epochs. We did not find consistent correlation between $\varepsilon$ and model performance, although this is tied to $M$ inherently (expecting reduced variance with higher $M$). With fixed $\varepsilon$, greater $M$ led to higher performance, as we can evaluate more models and identify the "correct" model-client assignments earlier on.

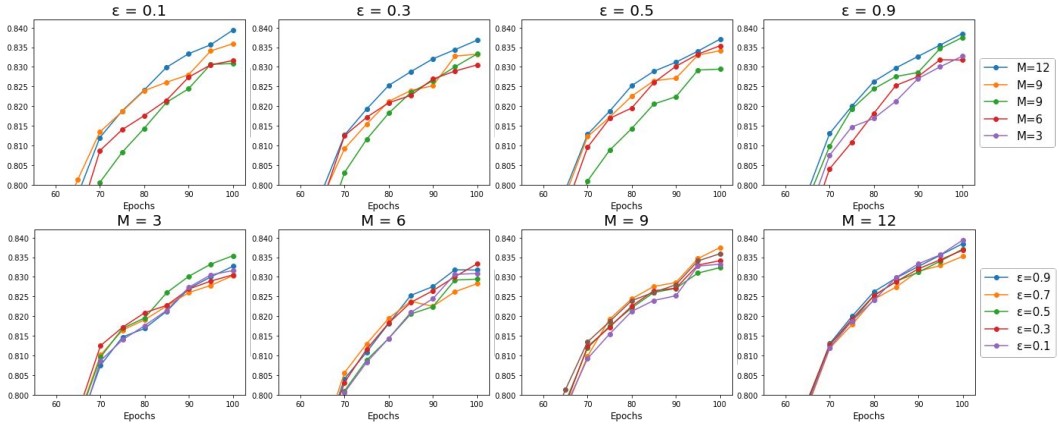

Figure 3: Ablations over $\varepsilon$-greedy exploration and number of models downloaded on CIFAR-10.

**Out-of-local-distribution personalization** We now consider the non-IID federated setting where each client optimizes for target distributions not the same as their local data distribution. Here, although a client may sufficiently train an adequate model for one domain, it has another target data distribution of interest with hard to access relevant data. For example, in a self-driving scenario, a client may not have enough data for certain classes due to geographical constraints, motivating the need to leverage info from others. To simulate this scenario, after organizing data into latent distributions, we randomly shuffle $(\boldsymbol{D}^{\text{val}}, \boldsymbol{D}^{\text{test}})$ as a pair among clients. We test on the CIFAR-10 and CIFAR-100 datasets with 15 clients, full participation, and 5 latent distributions, repeating the shuffling five times, and report mean accuracy over all clients.

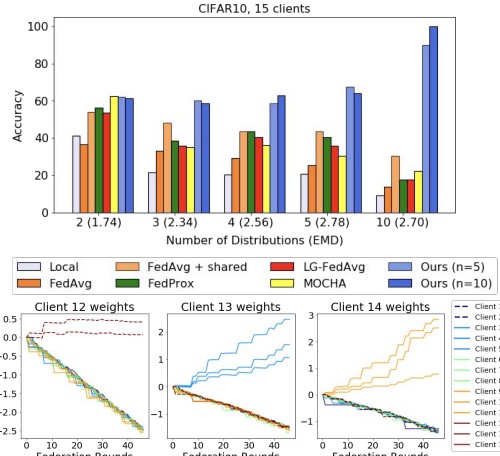

| | CIFAR-10 | CIFAR-100 |
|---|---|---|
| Local Training | $20.39 \pm 3.36$ | $7.40 \pm 1.31$ |
| FedAvg | $23.11 \pm 2.51$ | $13.06 \pm 1.48$ |
| FedAvg + Data | $42.15 \pm 2.16$ | $24.98 \pm 4.98$ |
| FedProx | $39.79 \pm 8.57$ | $14.39 \pm 2.85$ |
| LG-FedAvg | $38.95 \pm 1.85$ | $18.50 \pm 1.10$ |
| MOCHA | $30.80 \pm 2.60$ | $13.73 \pm 2.83$ |
| Clustered FL | $29.73 \pm 3.67$ | $19.75 \pm 1.58$ |
| Per-FedAvg | $39.8 \pm 5.38$ | $21.30 \pm 1.35$ |
| pFedMe | $43.7 \pm 7.27$ | $25.41 \pm 2.33$ |
| Ours (n=5) | $\mathbf{64.06 \pm 2.80}$ | $34.43 \pm 1.48$ |
| Ours (n=10) | $63.98 \pm 1.81$ | $\mathbf{40.94 \pm 1.62}$ |

Figure 4: **Top** Personalization on target distribution $\neq$ that of local training data. **Bottom** FedFomo upweights other clients with local data $\sim$ target distribution (5 latent non-IID dist.)

Table 3: Out-of-client distribution evaluation with 5 latent distributions and 15 clients. FedFomo outperforms all alternatives in various datasets.

As shown in Fig. 4 and Table 3, our method consistently strongly outperforms alternatives in both non-IID CIFAR-10 and CIFAR-100 federated settings. We compare methods using the same train and test splits randomly shuffled between clients, such that through shuffling we encounter potentially large amounts of data variation between a client's training data and its test set. This then supports the validity of the validation split and downloaded model evaluation components in our method to uniquely optimize for arbitrary data distributions different from a client's local training data. All methods other than ours are unable to convincingly handle optimizing for a target distribution that is different from the client's initially assigned local training data. Sharing data expectedly stands out among other methods that do not directly optimize for a client's objective, as each client then increases the label representation overlap between its train and test sets. We note that in the 2-distribution setting, where each client's training data consists of 5 classes on average, the higher performance of other methods may likely be a result of our simulation, where with only two distributions to shuffle between it is more likely that more clients end up with the same test distribution.

To shed further light on FedFomo's performance, we visualize how client weights evolve over time in this setting (Fig. 4 bottom), where to effectively personalize for one client, FedFomo should specifically increase the weights for the other clients belonging to the original client's target distribution. Furthermore, in the optimal scenario we should upweight *all* clients with this distribution while downweighting the rest. Here we show that this indeed seems to be the case, denoting local training distributions with color. We depict clients 12, 13, and 14, which all carry the same local data distribution, but 13 and 14 optimize for out-of-local distributions. In all cases, FedFomo upweights clients specifically carrying the same data distribution, such that while with shuffling we do not know apriori 13 and 14's target distributions, FedFomo discovers these and who should federate with whom in this setting as well. We include similar plots for all clients in Appendix A.2 (Fig. 6).

**Locally Private FedFomo** While we can implement FedFomo such that downloaded model parameters are inaccessible and any identifying connections between clients and their uploaded models are removed to subsequently preserve anonymity, unique real world privacy concerns may rise when sharing individual model parameters. Accordingly, we now address training FedFomo under $(\varepsilon, \delta)$-differential privacy (DP). Dwork et al. (2014) present further details, but briefly DP ensures that given two near identical datasets, the probability that querying one produces a result is nearly the same as querying the other (under control by $\varepsilon$ and $\delta$). Particularly useful here are DP's composability and robustness to post-processing, which ensure that if we train model parameters $\theta$ to satisfy DP, then any function on $\theta$ is also DP. We then perform local training with DP-SGD (Abadi et al., 2016) for a DP variant of FedFomo, which adds a tunable amount of Gaussian noise to each gradient and reduces the connection between a model update and individual samples in the local training

| Method | $\delta$ | $\sigma$ | CIFAR-10 | | CIFAR-100 | |
|---|---|---|---|---|---|---|
| | | | $\varepsilon$ | Accuracy | $\varepsilon$ | Accuracy |
| FedAvg | $1 \times 10^{-5}$ | 0 | $\infty$ | $19.37 \pm 1.42$ | $\infty$ | $5.09 \pm 0.38$ |
| FedAvg | $1 \times 10^{-5}$ | 1 | $10.26 \pm 0.21$ | $17.60 \pm 1.64$ | $8.20 \pm 0.69$ | $5.05 \pm 0.31$ |
| FedAvg | $1 \times 10^{-5}$ | 2 | $3.57 \pm 0.08$ | $16.19 \pm 1.62$ | $2.33 \pm 0.21$ | $4.33 \pm 0.27$ |
| Ours | $1 \times 10^{-5}$ | 0 | $\infty$ | $71.56 \pm 1.20$ | $\infty$ | $26.76 \pm 1.20$ |
| Ours | $1 \times 10^{-5}$ | 1 | $\mathbf{6.89 \pm 0.13}$ | $\mathbf{71.28 \pm 1.06}$ | $\mathbf{8.20 \pm 0.69}$ | $\mathbf{26.14 \pm 1.05}$ |
| Ours | $1 \times 10^{-5}$ | 2 | $1.70 \pm 0.04$ | $65.97 \pm 0.95$ | $1.71 \pm 0.15$ | $15.95 \pm 0.94$ |

Table 4: In-distribution classification with differentially private federated learning. With DP-SGD, `FedFomo` maintains high personalization accuracy with reasonable privacy guarantees.

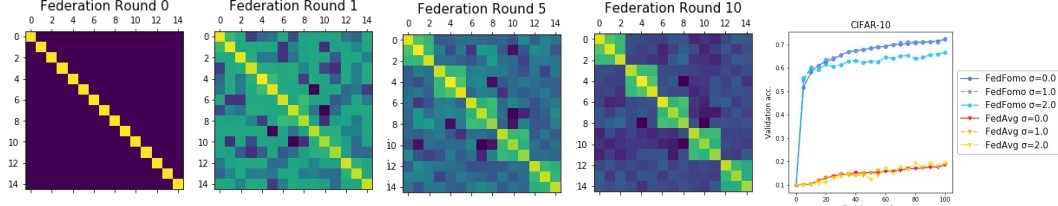

Figure 5: **Left**: Even with privacy-preserving updates, `FedFomo` still uncovers the underlying data distributions at large. **Right** We gain privacy benefits without substantial drop in performance.

data. More noise makes models more private at the cost of performance, and here we investigate if `FedFomo` retains its performance with increased privacy under noisy local updates.

We consider the in-distribution personalization task with 5 latent non-IID distributions from the CIFAR-10 and CIFAR-100 datasets, with 15 clients and full participation at each round, and compare `FedFomo` against `FedAvg` with varying levels of Gaussian noise, specified by $\sigma$. With all other parameters fixed, higher $\sigma$ should enable more noisy updates and greater privacy (lower $\varepsilon$), at the potential cost of performance. At fixed $\delta$, we wish to obtain high classification accuracy and low $\varepsilon$. We use the Opacus Pytorch library[2] for DP-SGD, and as baselines run `FedFomo` and `FedAvg` with the library's provided SGD optimizer with $\sigma = 0$. For DP runs, we set $\delta = 1 \times 10^{-5} \ll 3 \times 10^{-4}$, the inverse of the average number of local data points of each client, to maintain reasonable privacy.

In Table 4, `FedFomo` is able to retain a sizeable improvement over `FedAvg`, even against the non-DP `FedAvg`, and does so with minimal $\varepsilon$. As expected, greater $\sigma$ leads to improved privacy (lower $\varepsilon$) at the cost of decreased performance. Additionally, in Fig. 5 we show that even with noisy gradients to protect individual data point privacy, `FedFomo` maintains its ability to discover the larger latent distributions among local data (albeit with more noise initially). Most importantly, despite adding noise that could potentially derail our federated update, we are able to substantially reduce privacy violation risks under $(\varepsilon, \delta)$-differential privacy while maintaining strong performance.

## 5 CONCLUSION

We present `FedFomo`, a flexible personalized FL framework that achieves strong performance across various non-IID settings, and uniquely enables clients to also optimize for target distributions distinct from their local training data. To do so, we capture the intuition that clients should download personalized weighted combinations of other models based on how suitable they are towards the client's own target objective, and propose a method to efficiently calculate such optimal combinations by downloading individual models in lieu of previously used model averages. Beyond outperforming alternative personalized FL methods, we empirically show that `FedFomo` is able to discover the underlying local client data distributions, and for each client specifically upweights the other models trained on data most aligned to the client's target objective. We finally explore how our method behaves with additional privacy guarantees, and show that we can still preserve the core functionality of `FedFomo` and maintain strong personalization in federated settings.

---

[2]github.com/pytorch/opacus

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

# A APPENDIX

## A.1 DERIVING THE FOMO UPDATE

Recall that we can view each federated model download can be viewed as an iterative update,

$$\theta_i^{\ell(t+1)} = \theta_i^{\ell(t)} + \sum_{n=1}^{N} w_n \cdot \left(\theta_n^{\ell(t)} - \theta_i^{\ell(t)}\right) \tag{4}$$

where given a client's current parameters $\theta_i^{\ell(t)}$, the weights $\boldsymbol{w} = \langle w_1, \ldots, w_N \rangle$ in conjunction with the model deltas $\theta_n^{\ell(t)} - \theta_i^{\ell(t)}$ determine how much each client should move its local model parameters to optimize for some objective. Unlike more common methods in machine learning such as gradient descent, the paths we can take to get to this objective are restricted by the fixed model parameters $\{\theta_n^{\ell(t)}\}$ available to us at time $t$. While traditional FL methods presume this objective to be global test set performance, from a client-centric perspective we should be able to set this objective with respect to any dataset or target distribution of interest.

We then view this problem as a constrained optimization problem where $\sum_{n \in [N]} w_n = 1$. As a small discrepancy, if $i \in [N]$, then to also calculate $w_i$ or how much client $c_i$ should weigh its own model in the federated update directly, we reparameterize Eq. 4 as an update from a version of the local model *prior* to its current state, e.g.

$$\theta_i^{\ell(t+1)} = \theta_i^{\ell(t-1)} + \sum_{n=1}^{N} w_n \cdot \left(\theta_n^{\ell(t)} - \theta_i^{\ell(t-1)}\right) \tag{5}$$

and again we have current budget of 1 to allocate to all weights $\boldsymbol{w}$. Additionally, to go along with Eq. 5, we deviate a bit from the optimal $t + 1$ term in Eq. 2 and set

$$\theta_i^{\ell(t+1)} = \theta_i^{\ell(t)} \leftarrow \theta_i^{\ell(t-1)} - \alpha \mathbf{1}^{\top} \nabla_{\boldsymbol{w}} \mathcal{L}_i(\theta_i^{\ell(t-1)}) \tag{6}$$

There is then a parallel structure between Eq. 5 and Eq. 6, and we proceed by trying to find optimal $\boldsymbol{w}$ that would let our update in Eq. 6 closely approximate the optimal update taking the gradient $\nabla_{\boldsymbol{w}}$. We accordingly note the equivalence from Eq. 5 and Eq. 6, where for desired $w_n$,

$$\sum_{n=1}^{N} w_n \cdot \left(\theta_n^{\ell(t)} - \theta_i^{\ell(t-1)}\right) = -\alpha \mathbf{1}^{\top} \nabla_{\boldsymbol{w}} \mathcal{L}_i(\theta_i^{\ell(t-1)}) \tag{7}$$

or in matrix form:

$$\begin{bmatrix} w_1 \\ \vdots \\ w_N \end{bmatrix}^{\top} \begin{bmatrix} (\theta_1^{\ell(t)} - \theta_i^{\ell(t-1)}) \\ \vdots \\ (\theta_N^{\ell(t)} - \theta_i^{\ell(t-1)}) \end{bmatrix} = \begin{bmatrix} -\alpha \\ \vdots \\ -\alpha \end{bmatrix}^{\top} \begin{bmatrix} \frac{\partial}{\partial w_1} \mathcal{L}_i(\theta_i^{\ell(t-1)}) \\ \vdots \\ \frac{\partial}{\partial w_N} \mathcal{L}_i(\theta_i^{\ell(t-1)}) \end{bmatrix} \tag{8}$$

Then for each weight $w_n$, we solve for its optimal value by equating the left and right-hand corresponding vector components. We do so by deriving a first order approximation of $\frac{\partial}{\partial w_n} \mathcal{L}_i(\theta_i^{\ell(t-1)})$. First, for each $w_n$, we define the function:

$$\varphi_n(w) := w_n \cdot \theta_n^{\ell(t)} + (1 - w_n) \cdot \theta_i^{\ell(t-1)} \tag{9}$$

as an alternate parameterization of the $\theta$'s as functions of weights. We can see that for all $n \in [N]$,

$$\varphi_n(0) = \theta_i^{\ell(t-1)}$$

$$\Rightarrow \frac{\partial}{\partial w_n} \mathcal{L}_i(\theta_i^{\ell(t-1)}) = \frac{\partial}{\partial w_n} \mathcal{L}_i(\varphi_n(0))$$

Then using a first-order Taylor series approximation, we also note that

$$\mathcal{L}_i(\varphi_n(w')) \approx \mathcal{L}_i(\varphi_n(0)) + \frac{\partial}{\partial w_n}\mathcal{L}_i(\varphi_n(0))(w' - 0) \tag{10}$$

such that at our initial point $w = 0$ or $\theta_i^{\ell(t-1)}$, we can approximate the derivative $\frac{\partial}{\partial w_n}\mathcal{L}_i(\varphi_n(0))$ when $w' = 1$ as:

$$\frac{\partial}{\partial w_n}\mathcal{L}_i(\varphi_n(0)) = \mathcal{L}_i(\varphi_n(1)) - \mathcal{L}_i(\varphi_n(0))$$

$$\Rightarrow \frac{\partial}{\partial w_n}\mathcal{L}_i(\theta_i^{\ell(t-1)}) = \mathcal{L}_i(\theta_n^{\ell(t)}) - \mathcal{L}_i(\theta_i^{\ell(t-1)}) \tag{11}$$

following from Eq. 9. Then for each vector element in Eq. 8, indexed by $n = [N]$, we can plug in the corresponding partial derivative from Eq. 11 and solve for the corresponding $w_n$ to get

$$w_n = -\alpha \cdot \frac{\mathcal{L}_i(\theta_n^{\ell(t)}) - \mathcal{L}_i(\theta_i^{\ell(t-1)})}{\|\theta_n^{\ell(t)} - \theta_i^{\ell(t-1)}\|} \tag{12}$$

as the individual weight for client $c_i$ to weight model $\theta_n$ in its federated update.

We arrive at Eq. 3 by distributing the negative $\alpha$ to capture the right direction in each update, but also note that the constant cancels out because we normalize to ensure our weights sum to 1, such that the weights $w_n^*$ that we actually use in practice are given by:

$$w_n^* = \frac{\max(w_n, 0)}{\sum_{n=1}^{N} \max(w_n, 0)} \tag{13}$$

## A.2 ADDITIONAL LATENT DISTRIBUTION NON-IID EXPERIMENTS

**CIFAR-100**  Here we show results on the latent non-IID in-distribution personalization setup for the CIFAR-100 dataset. As in the CIFAR-10 setting, we compare `FedFomo` against various recent alternative methods when personalizing to a target distribution that is the same as the client's local training data, and report accuracy as an average over all client runs. We also show results partitioning the CIFAR-100 dataset into increasing number of data distributions for 15 clients total, and report the increasing EMD in parentheses. In Table 5, `FedFomo` consistently outperforms all alternatives with more non-IID data across different clients. We note similar patterns to that of the CIFAR-10 dataset, where our method is more competitive when client data is more similar (lower EMD, number of distributions), but handily outperforms others as we increase this statistical label heterogeneity.

| | CIFAR-100 Number of Latent Distributions (EMD) | | | | |
|---|---|---|---|---|---|
| Method | 2 (1.58) | 3 (1.96) | 4 (2.21) | 5 (2.41) | 10 (2.71) |
| Local Training | $23.36 \pm 0.33$ | $23.89 \pm 2.03$ | $28.44 \pm 1.97$ | $23.11 \pm 9.44$ | $41.26 \pm 1.31$ |
| FedAvg | $28.01 \pm 0.74$ | $18.95 \pm 0.22$ | $25.69 \pm 0.41$ | $21.26 \pm 0.89$ | $18.19 \pm 0.987$ |
| FedAvg + Data | $28.12 \pm 0.63$ | $19.01 \pm 0.27$ | $25.85 \pm 0.27$ | $25.18 \pm 0.39$ | $18.23 \pm 0.835$ |
| FedProx | $28.21 \pm 0.79$ | $27.78 \pm 0.000$ | $25.79 \pm 0.05$ | $24.93 \pm 0.38$ | $18.18 \pm 0.82$ |
| LG-FedAvg | $26.97 \pm 7.52$ | $24.69 \pm 4.29$ | $24.79 \pm 4.50$ | $25.62 \pm 5.70$ | $27.53 \pm 9.12$ |
| MOCHA | $33.66 \pm 4.14$ | $33.61 \pm 7.88$ | $29.44 \pm 8.30$ | $32.34 \pm 7.09$ | $34.72 \pm 7.80$ |
| Clustered FL | $\mathbf{41.50 \pm 6.66}$ | $36.36 \pm 10.72$ | $37.41 \pm 8.30$ | $36.78 \pm 12.05$ | $34.43 \pm 10.14$ |
| Per-FedAvg | $32.14 \pm 6.90$ | $32.22 \pm 7.37$ | $34.50 \pm 5.81$ | $36.58 \pm 6.72$ | $38.41 \pm 7.41$ |
| pFedMe | $31.53 \pm 3.83$ | $32.39 \pm 5.36$ | $30.86 \pm 3.75$ | $30.86 \pm 3.80$ | $37.70 \pm 2.13$ |
| Ours (n=5) | $35.44 \pm 1.91$ | $36.21 \pm 4.92$ | $38.41 \pm 2.58$ | $42.96 \pm 1.24$ | $\mathbf{44.29 \pm 1.22}$ |
| Ours (n=10) | $37.09 \pm 1.95$ | $\mathbf{37.09 \pm 3.84}$ | $\mathbf{39.94 \pm 0.74}$ | $\mathbf{43.06 \pm 0.42}$ | $43.75 \pm 1.74$ |

Table 5: In-distribution personalized federated classification on the CIFAR-100 dataset

### A.3 CLIENT WEIGHTING WITH PERSONALIZATION

**In-local vs out-of-local distribution personalization** Following the visualizations for client weights in the out-of-local distribution personalization setting (Fig. 4), we include additional visualizations for the remaining clients (Fig. 6). For comparison, we also include the same visualizations for the 15 client 5 non-IID latent distribution setup on CIFAR-10, but when clients optimize for a target distribution the same as their local training data's (Fig. 7). In both, we use color to denote the client's local training data distribution, such that if `FedFomo` is able to identify the right clients to federated with that client, we should see the weights for those colors increase or remain steady over federation rounds, while all other client weights drop.

As seen in both Fig. 6 and Fig. 7, `FedFomo` quickly downweights clients with unhelpful data distributions. For the in-distribution personalization, it is able to increase and maintain higher weights for the clients from the same distribution, and consistently does so for the other two clients that belong to its distribution. In the out-of-local distribution personalization setting, due to our shuffling procedure we have instances where certain clients have in-distribution targets, while others have out-of-distribution targets. We see that `FedFomo` is able to accommodate both simultaneously, and learns to separate all clients belonging to the target distributions of each client from the rest.

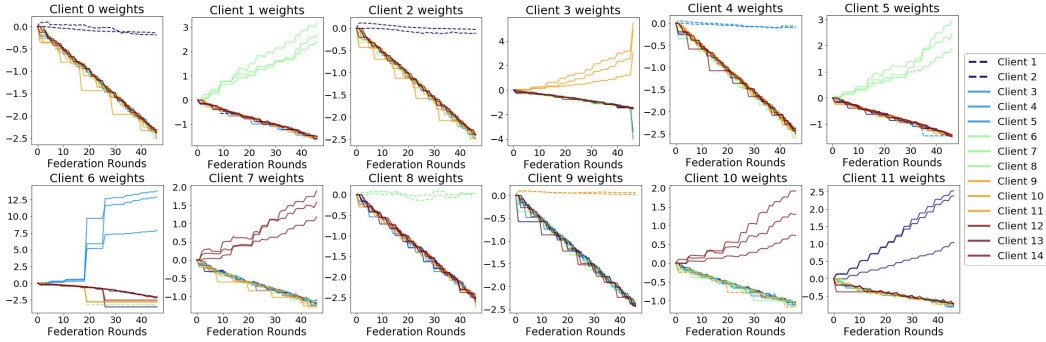

Figure 6: Client-to-client weights over time when personalizing for non-local target distributions. `FedFomo` quickly downweights non-relevant clients while upweighting those that are helpful.

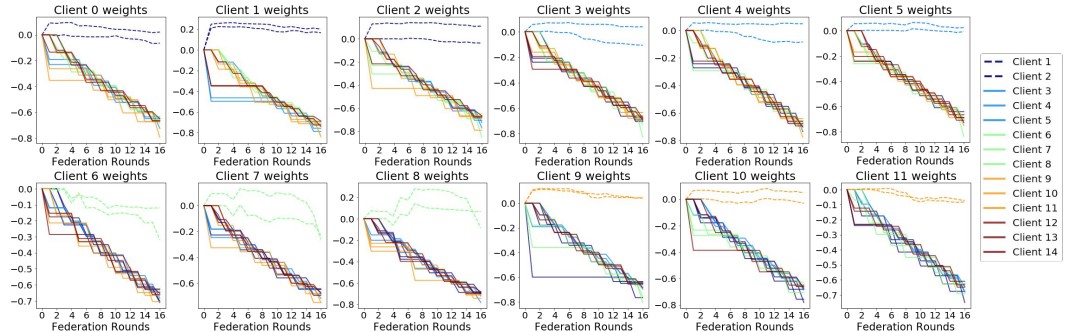

Figure 7: Client-to-client weights over time when personalizing for local target distributions. `FedFomo` downweights non-relevant clients while upweighting or keeping steady helpful ones.

| Method | $\delta$ | $\sigma$ | CIFAR-10 | | CIFAR-100 | |
|--------|----------|----------|----------|----------|-----------|----------|
| | | | $\varepsilon$ | Accuracy | $\varepsilon$ | Accuracy |
| FedAvg | $1 \times 10^{-5}$ | 0 | $\infty$ | $19.37 \pm 1.42$ | $\infty$ | $5.09 \pm 0.38$ |
| FedAvg | $1 \times 10^{-5}$ | 1 | $10.26 \pm 0.21$ | $17.60 \pm 1.64$ | $8.20 \pm 0.69$ | $5.05 \pm 0.31$ |
| FedAvg | $1 \times 10^{-5}$ | 2 | $3.57 \pm 0.08$ | $16.19 \pm 1.62$ | $2.33 \pm 0.21$ | $4.33 \pm 0.27$ |
| Ours | $1 \times 10^{-5}$ | 0 | $\infty$ | $71.56 \pm 1.20$ | $\infty$ | $26.76 \pm 1.20$ |
| Ours | $1 \times 10^{-5}$ | 1 | $\mathbf{6.89 \pm 0.13}$ | $\mathbf{71.28 \pm 1.06}$ | $\mathbf{8.20 \pm 0.69}$ | $\mathbf{26.14 \pm 1.05}$ |
| Ours | $1 \times 10^{-5}$ | 2 | $1.70 \pm 0.04$ | $65.97 \pm 0.95$ | $1.71 \pm 0.15$ | $15.95 \pm 0.94$ |
| Ours (MA) | $1 \times 10^{-5}$ | 0 | $\infty$ | $47.90 \pm 2.79$ | $\infty$ | $12.02 \pm 1.34$ |
| Ours (MA) | $1 \times 10^{-5}$ | 1 | $9.26 \pm 0.19$ | $46.33 \pm 4.04$ | $10.32 \pm 0.89$ | $11.60 \pm 0.65$ |
| Ours (MA) | $1 \times 10^{-5}$ | 2 | $3.20 \pm 0.07$ | $43.76 \pm 3.08$ | $3.22 \pm 0.28$ | $9.52 \pm 0.72$ |

Table 6: In-distribution classification with differentially private federated learning, with the addition of `FedFomo` with a model average baseline (Ours (MA)).

## A.4 Additional Privacy Experiments

As a follow-up on the privacy experiments in Section 4, we also consider a multiple model variant of `FedFomo`, where instead of a client downloading a single model $\theta_n$ and evaluating against its own previous model $\theta_i^{t-1}$, the client downloads the simple average of all the uploaded models *except* $\theta_n$ (i.e. $\frac{1}{N-1} \sum_{j \in [N] \setminus n} \theta_n$) and compares this against the simple average of all uploaded models. This tackles an orthogonal notion of privacy compared to the previous solution of introducing noise to local model gradients via DP-SGD, as now individual data point membership is harder to distill from shared parameters that come from the average of multiple local models. To calculate weights, we note a sign change with respect to Eq. 3 and the baseline model, as now $w_n$ should be positive if the model average without $\theta_n$'s contribution results in a larger target objective loss than the model average with $\theta_n$. Given client $c_i$ considering model $\theta_n$, this leads to `FedFomo` weights:

$$w_n \propto \mathcal{L}_i\Big(\frac{1}{N-1} \sum_{j \in [N] \setminus n} \theta_j\Big) - \mathcal{L}_i\Big(\frac{1}{N} \sum_{j \in [N]} \theta_j\Big) \tag{14}$$

We evaluate this variant with the same comparison over $(\varepsilon, \delta)$-differential privacy parameters on the 15 client 5 latent-distribution scenarios in our previous privacy analysis. We set $\delta = 1 \times 10^{-5}$ to setup practical privacy guarantees with respect to the number of datapoints in each client's local training set, and consider Gaussian noise $\sigma \in \{0, 1, 2\}$ for baseline and $(\varepsilon, \delta)$-differentially private performances. At fixed $\delta$, we wish to obtain high classification accuracy with low privacy loss ($\varepsilon$).

In Table 6 we include results for this model average baseline variant (Ours (MA)) on the CIFAR-10 and CIFAR-100 datasets, along with the differentially private federated classification results in Table 4 using DP-SGD during local training for additional context. For both datasets, we still handily outperform non-private `FedAvg`, although performance drops considerably with respect to the single model download `FedFomo` variant. We currently hypothesize that this may be due to a more noisy calculation of another model's potential contribution to the client's current model, as we now consider the effects of many more models in our loss comparisons as well. Figuring out a balance between the two presented weighting schemas to attain high personalization and high privacy by downloading model averages then remains interesting future work.

## A.5 LATENT DISTRIBUTION NON-IID MOTIVATION AND SETUP

In this subsection, we discuss our latent distribution non-IID setting in more detail. We believe the pathological setup though useful might not represent more realistic or frequent occurring setups. As an example, a world-wide dataset of road landscapes may vary greatly across different data points, but variance in their feature representations can commonly be explained by their location. In another scenario, we can imagine that certain combinations of songs, or genres of music altogether are more likely to be liked by the same person than others. In fact, the very basis and success of popular recommender system algorithms such as collaborative filtering and latent factor models rely on this scenario (Hofmann, 2004). Accordingly, in this sense statistical heterogeneity and client local data non-IIDnes is more likely to happen in groups.

We thus propose and utilize a latent distribution method to evaluate `FedFomo` against other more recent proposed FL work. To use this setting, we first compute image representations by training a VGG-11 convolutional neural network to at least 85% classification accuracy on a corresponding dataset. We then run inference on every data point, and treat the 4096-dimensional vector produced in the second fully-connected layer as a semantic embedding for each individual image. After further reduction to 256 dimensions through PCA, we use K-Means clustering to partition our dataset into $D$ disjoint distributions. Given $K$ total clients, we then evenly assign each client to a distribution $\mathcal{D}$. For each client we finally obtain its local data by sampling randomly from $\mathcal{D}$ without replacement. For datasets with pre-defined train and test splits, we cluster embeddings from both at the same time such that similar images across splits are assigned the same K-means cluster, and respect these original splits such that all $\boldsymbol{D}^{\text{test}}$ images come from the original test split. (Fig. 8)

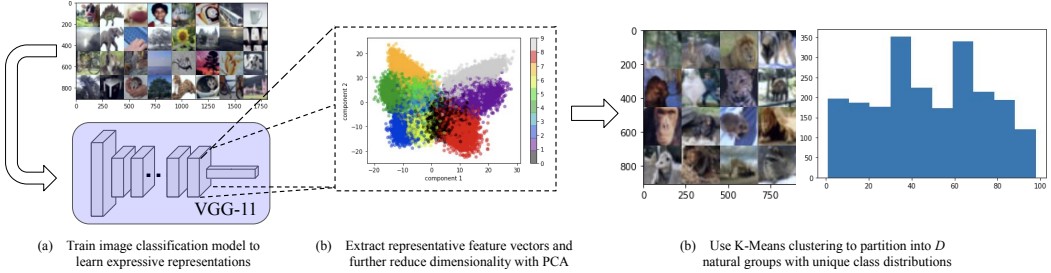

(a)  Train image classification model to learn expressive representations
(b)  Extract representative feature vectors and further reduce dimensionality with PCA
(b)  Use K-Means clustering to partition into $D$ natural groups with unique class distributions

Figure 8: Visual overview for generating latent distributions using image classification datasets.

## A.6 MODEL IMPLEMENTATION DETAILS

We train with SGD, 0.1 learning rate, 0 momentum, 1e-4 weight decay, and 0.99 learning rate decay for CIFAR-10/100, and do the same except with 0.01 learning rate for MNIST. For `FedFomo` we use $n = 5$ and $n = 10$ downloads per client, $\varepsilon = 0.3$ with 0.05 decay each round, and separate $\boldsymbol{D}^{\text{train}}$ and $\boldsymbol{D}^{\text{val}}$ with an 80-20 split.

## A.7 ADDITIONAL DESIGN ABLATIONS

In this section we present additional work on key hyperparameters or aspects of `FedFomo` to give further insight into our method's functionality and robustness to parameters. We consider key design choices related to the size of each client's validation split.

**Size of the validation split**  To better organize federated uploaded models into personalized federated updates, our method requires a local validation split $\boldsymbol{D}^{\text{val}}$ that reflects the client's objective or target test distribution. Here, given a pre-defined amount of locally available data, we ask the natural question of how a client should best go about dividing its data points between those to train its own local model and those to evaluate others with respect to computing a more informed personalized update through `FedFomo`. We use the 15 client 100% participation setup with 5 latent distributions organized over the CIFAR-10 dataset, and consider both the evaluation curve and final test accuracy over allocating a fraction $\in \{0.01, 0.05, 0.1, 0.2, 0.4, 0.6, 0.8, 0.9\}$ of all clients' local data to $\boldsymbol{D}^{\text{val}}$, and track evaluation over 20 communication rounds with 5 epochs of local training per round. On

average, each client has 3333 local data points. We denote final accuracy and standard deviation over five runs in Fig 9.

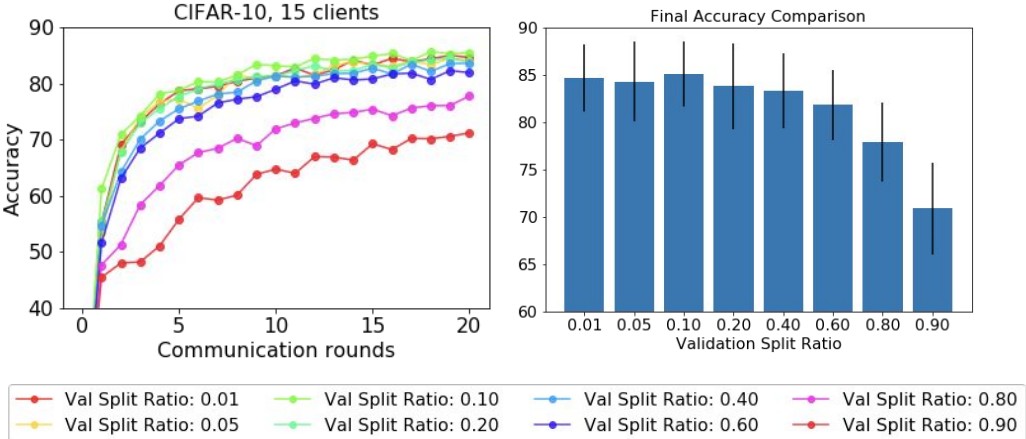

Figure 9: In-distribution accuracy over validation split ratio.

As reported in Fig. 9, we observe faster convergence to a higher accuracy when allocating under half of all local data points to the validation split, with a notable drop-off using more data points. This is most likely a result of reducing the amount of data available for each client to train their model locally. Eventually this stagnates, and observe a slight decrease in performance between validation split fraction 0.05 and 0.1.

