# OpenReview forum: "Personalized Federated Learning with First Order Model Optimization"
_ICLR.cc/2021/Conference — ICLR 2021 Poster_

### Official Review · AnonReviewer4 · 2020-10-22
**The paper addresses an important topic in federated learning which is personalization. However, more experiments/discussion to be added in regard to comparison with other methods.**

**Rating:** 7
**Confidence:** 3

**Review:**

**Paper Summary:** The paper addresses an important topic in federated learning which is personalization. The authors propose a two steps process to achieve the personalization: 1. Figuring out which models to send to which clients; 2. Computing their personalized weighted combinations for each client. To determine the weights, the authors use first order approximation.

**Pros**:
1. Personalized federated learning is an important problem, and I see a lot of value in practical applications of federated learning
2. The proposed method is simple and intuitive
**Cons**:
1. I see strong claims in the paper such as 'we are the first to our knowledge to enable transfer to new specific data domains of interest through personalized FL. '
2. Some references are missing in the related work sections such as (Dinh et al. NeurIPS 2020), (Fallah et al. ArXiv 2020), and (Peterson et al. ArXiv 2019).

**Areas to Improve**:
1. I like to see more experiments to be added in regards to comparison with some recent works especially with (Dinh et al. NeurIPS 2020).
2. Would be helpful to provide a convergence analysis of the proposed algorithm.

**Missing References:**
1. Personalized Federated Learning with Moreau Envelopes (Dinh et al. NeurIPS 2020)
2. Private Federated Learning with Domain Adaptation (Peterson et al. ArXiv 2019)
3. Personalized Federated Learning: A Meta-Learning Approach (Fallah et al. ArXiv 2020)

**Minor Concerns:**
1. Grammatical error here 'given a local objective clients their received models and'
2. Eq. 6 some errors in the superscript of "\theta"

---

> ### Author Response · Authors · 2020-11-21
> **Added experiments and further insights comparing to requested methods, clarified claims**
>
> Thank you for your review on our “simple and intuitive” approach for personalized federated learning. We appreciate your constructive comments and following your suggestions  added results  for both (Dinh et al. NeurIPS 2020) and (Fallah et al. NeurIPS 2020) in our main evaluation settings (Table 1, Table 2, Table 3, Table 5). We also added in the missing references and corrected for the minor edits.
>
> **Experiments with more recent work** Table 1 now shows that our method is still able to outperform more recent work such as Per-FedAvg (Fallah et al. NeurIPS 2020)  or pFedMe  (Dinh et al. NeurIPS 2020). Meanwhile in Table 2 for various non-IID latent distribution setups FedFomo also outperforms these alternatives, although the results are more competitive with greater non-IIDnes. Interestingly, in the out-of-local distribution setting, while FedFomo still outperforms these methods (with greater insight into why given by inspecting the client-to-client weights in Fig. 4, Fig. 7) we see that the requested methods achieve stronger performance than previous comparisons. We think there is more to explore regarding the personalization capabilities of all these methods, and thank the reviewer for their suggestion as we believe further comparing different notions of personalized adaptation is a worthwhile pursuit and well-motivated suggestion to more deeply investigate in future work.
>
> **Strong claims** After reviewing related works including both suggested works, we thank the reviewer for pointing this out and can understand where we think this is coming from. Technically, work such as Dinh et al. 2020 also does this because they deal with transfer from a global model to a personalized one, where from the global server’s perspective any participating client’s local data can be a specific data domain of interest. What we mean to say here and what we argue still stands is that from the client’s perspective, we are the first to show how FL approach can enable personalization to a new domain other than that of the client’s local training data. We believe this highlights an important practical use case of federated learning, where although a client may be able to reasonably locally train a model for one domain, it chooses to collaborate because it is specifically interested in obtaining a stronger model for another domain of interest it is currently weak in, e.g. from not having direct access to enough data points. We believe so far that this notion has not been explored in the literature, whereas we provide a simple solution that naturally accommodates for it. We are reviewing the current manuscript again however for instances like this to clarify such claims.
>
> **Convergence Analysis** We agree that a convergence analysis could lend useful insights and thank you for this suggestion. However, in this revision we focused on implementing and adding in the requested more recent works, and including additional empirical evidence to give further insight into our method and reasons behind why we outperform other methods. For example figures 2 and 5 show that across a variety of federated settings (different number of clients, different levels of non-IIDness, training with differential privacy), FedFomo is able to empirically converge to finding the “right”  underlying distributions behind local client data. We appreciate the feedback on this however and agree that there is value to formalizing this empirical evidence.

---

> > ### Comment · AnonReviewer4 · 2020-11-24
> > **Paper is improved after revision**
> >
> > Given that the paper has been improved after the revision I have raised my score. I also highly recommend to open source the code for reproducibility purposes.

---

### Official Review · AnonReviewer3 · 2020-10-25
**Interesting view of FL personalization, with some concerns**

**Rating:** 6
**Confidence:** 3

**Review:**

This paper proposed FOMO, a personalized FL framework. Comparing to existing methods that conduct local adaptation starting from a single global model, FOMO keeps track of all models, and personalize by computing the weighted version among all models.  The experimental results show improvement comparing to existing methods.

The following points from this paper are novel and interesting:

(1) personalization via weighting multiple models is intuitively more stable, comparing to client-side gradient descent. Especially for non-iid settings, personalize from a set of models, offers more flexibility and robustness to personalize, than personalize from a single model. Moreover, assuming having a validation dataset, the client with multiple models can have an easy way to choose an existing model.

(2) The method to approximate optimal w, is clean and shows interesting improvement. Although the experiments are not done on FL golden standards such as EMNIST/Shakespear, the experiments and reasonings are well presented and well conveyed.

However, there are a few points that need to be taken care of:

(1) FOMO assumes that the server keeps track of all clients' models,  this changes the classical assumption of FL: all model updates should be anonymous. This leads to privacy issues. The author addresses the privacy concern in section A.3, yet I am not satisfied. Here is the reason: (a) when any model is downloaded to the client, there is no scheme to avoid the client to analyze the model since all models share the same structure, only weights are different. With the other's model, a malicious client can do some harm, which needs differential privacy (DP) guarantees. (b) The DP approach adds noise to updated models, which could lead to a significant personalization performance drop, which hasn't been discussed in the paper, and DP is critical to ensure data privacy. In my opinion, FOMO introduces a powerful idea of downloading multiple models to improve personalization, yet introduces the problem of privacy.

(2) Comparing to keep track of models of all clients (I am not satisfied with the claim in A.3), further research, should consider keeping track of multiple models, with anonymous FL updates. The clients can use multiple models to update, while since the models are updated via anonymous gradients, then no privacy issue will arise.

I feel this is an interesting direction for FL personalization research, while the privacy issue is not properly addressed. I am expecting the authors to address the privacy part with some updated experiment. Given the above concerns, I set my score to 5. If the privacy issue is properly addressed I am glad to increase the score.

---

> ### Author Response · Authors · 2020-11-21
> **Added new section on FedFomo with differential privacy to address privacy concerns**
>
> Thank you for your review! We especially appreciate the notes on the novelty and interesting aspects of our paper with respect to the intuitively more stable method, increased flexibility and robustness to personalize, clean method with interesting improvement, and well presented and well conveyed experiments and reasonings. To address the reviewer's concerns about privacy, we introduced a new experimental section (Locally Private FedFomo, Sec 4)  with new experiments training our models locally with $(\varepsilon, \delta)$-differential privacy in the revision, and show that even with DP guarantees, FedFomo achieves satisfying personalization. We discuss this further below.
>
> We agree that there are important points regarding the privacy implications of FedFomo, and thank the reviewer for bringing further to light potential concerns and this inherent trade-off introduced in our method between privacy and personalization. We think this is an exciting area to explore further, and addressed this in a new section (Locally Private FedFomo, Sec 4). Inspired by the reviewer's suggestion, we incorporated DP-SGD (Abadi et al. 2016) in our local training procedure, arguing that because of DP's composability and robustness to post-processing, this is a natural incorporation that renders the downloaded model parameters secure in a DP sense. As noted by the reviewer, this added noise could reduce personalization performance, and we investigated the effects of these noisy gradient updates to protect individual data point across CIFAR-10 and CIFAR-100 non-IID federated data setups with reasonable $\delta$ and various Gaussian noise parameter $\sigma$.
>
> In table 4, we show that in non-IID settings for CIFAR-10 and CIFAR-100, our DP-variant of FedFomo still strongly outperforms FedAvg but with additional privacy guarantees. Meanwhile, we also show in Figure 5 that the added noise can be tuned such that we do not suffer from drastic personalization performance drops. Additionally, while the client-to-client weight updates are more noisy, we maintain the ability to correctly “discover” the clients’ local training data distributions at large under reasonable DP $\varepsilon$.
>
> We hope that these additions address the reviewer's concerns in a satisfying manner.
>
> **Further research** Based on the reviewer's comment to incorporate more models, we also considered an additional FedFomo weight calculation scheme to make our method more secure, in the sense that if clients always download model averages, then there it is harder for attackers to distill the individual model contributions. The basic premise of this is that instead of downloading a single model and comparing its loss against a local model to value its importance for a personalized update, we download a model average containing all models except that model, and compare the loss of it against the FedAvg baseline. We are currently working on incorporating this into the appendix of an updated revision, but show numbers on the 15 client CIFAR-10 5 non-IID latent distribution setting below (as Ours (MA)):
>
> | Method    | Delta    | Sigma | Epsilon        | Accuracy       |
> |-----------|----------|-------|----------------|----------------|
> | FedAvg    | 1.00E-05 |     0 | $\infty$       | $19.37 \pm 1.42$ |
> | FedAvg    | 1.00E-05 |     1 | $10.26 \pm 0.21$ | $17.60 \pm 1.64$ |
> | FedAvg    | 1.00E-05 |     2 | $3.57 \pm 0.08$  | $16.19 \pm 1.62$ |
> | Ours      | 1.00E-05 |     0 | $\infty$       | $71.56 \pm 1.20$ |
> | Ours      | 1.00E-05 |     1 | $6.89 \pm 0.13$  | $71.28 \pm 1.06$ |
> | Ours      | 1.00E-05 |     2 | $1.70 \pm 0.04$  | $65.97 \pm 0.95$ |
> |-----------|----------|-------|----------------|----------------|
> | Ours (MA) | 1.00E-05 |     0 | $\infty$       | $47.90 \pm 2.79$ |
> | Ours (MA) | 1.00E-05 |     1 | $9.26 \pm 0.19$  | $46.33 ± 4.04$   |
> | Ours (MA) | 1.00E-05 |     2 | $3.20 \pm 0.07$  | $43.76 \pm 3.08$ |
>
> We see that this formulation is also able to outperform FedAvg, albeit with decreased performance compared to the single download variant of FedFomo. We hope that this analysis can satisfyingly address and justify the potential privacy concerns brought up with our method. Again we think this is an interesting research consideration to pursue further, and are encouraged to conduct additional experiments in future work with additional insights into considerations such as robustness to both black and whitebox model inference attacks that may be particularly relevant to our method. However, here we hope to focus on the initial presentation of our method and its “powerful idea of downloading multiple models to improve personalization”.
>
> Abadi, M., Chu, A., Goodfellow, I., Mcmahan, H. B., Mironov, I., Talwar, K., &amp; Zhang, L. (2016). Deep Learning with Differential Privacy. Proceedings of the 2016 ACM SIGSAC Conference on Computer and Communications Security. https://doi.org/10.1145/2976749.2978318

---

### Official Review · AnonReviewer2 · 2020-10-28
**Interesting approach to personalized federated learning. Some design choices ask for more analysis**

**Rating:** 6
**Confidence:** 3

**Review:**

EDIT: Based on the author's modifications to the manuscript, I have increased my score from 5 to 6.

This paper introduces an approach to personalized federated learning. After each communication round, nodes receive full models from other nodes. The nodes construct a weighted average of the received models so that the mix performs well on their local *validation* data. The first interesting design choice is an approximation to the node-local optimal weighting problem. The second contribution is a scheme that decides which local models to share among which nodes, avoiding excessive $O(K^2)$ communication of full models. The proposed model is experimentally demonstrated to be competitive or better than previous work on an extensive array of tasks.

The writing is generally clear. I did have the feeling that the abstract, introduction and related work sections were slightly repetitive and could be made more concise.

The setup is partially motivated by data privacy. It does not require any data to be shared between users. It does, however, requiring many full models to be exchanged. While there might definitely be scenarios in which this is reasonable, the large amount of communication could be prohibitive in many practical scenarios, and exchanging local models might still raise privacy concerns.

Although I think the framework presented in this paper is interesting and potentially relevant, I do see shortcomings in the current manuscript:
- The two contributions (the approximation to the optimal weighting problem + model exchange coordiation scheme to save bandwidth) could be better motivated. For the weight computation, it is not completely clear to me why this is called 'first-order'. What is this a first order approximation of? What is the effect of this approximation compared to the 'real' solution or other choices you may have considered? For the model exchange coordination scheme, I am missing some details in the description. I am not confident I could implement this scheme from the paper. I would also expect to see some form of ablation study here: what is the effect of varying the number of other models shared with nodes. What is the effect of the parameter $\epsilon$, etc.'? From Table 1, it seems that sharing more models with workers (n=10) is not necessarily better than (n=5).
- There seems to be no explicit scheme to keep the personal models of the nodes close together. I would like to see some experimental or theoretical analysis or insight into why the models don't drift apart until they cannot benefit from each other anymore. How do the weights w_n evolve over time?


A few minor remarks:
- In Eqn. 5, I wonder if the denominator is correct. Is it dividing by a vector?
- Is division by zero never a problem in the proposed weighting normalization scheme?
- I couldn't follow the sentence "where clients (1) given a local objective clients their received models and".
- In "from some distribution $D_j$ and local" (notation section), I believe $D_j$ should be $D_i$.

---

> ### Author Response · Authors · 2020-11-21
> **Clarified motivations and details, added additional design ablations, further empirical insight into method (1/2)**
>
> Thank you for your careful and insightful review! We appreciate your comments pointing out our introduction of a new approach to personalized federated learning, with interesting design choices to avoid excess communication and approximate node-local optimal weights for personalization, which is competitive or better than previous work on an extensive array of tasks. We added a new experiment to address the comment on privacy (showing that our method works with differentially private local training) (Section 4.2). To address the other shortcomings, we also ran the requested ablation study and added these results to our paper (Fig 3), and collected new empirical evidence to report how the client-to-client weights and model parameters change over time (Figure 2, Figure 4), which we believe gives further support for our method. Finally we are grateful for the minor remarks, which we have edited in the revision along with a full pass to address the repetition.
>
> **Data Privacy**  We agree that there are naturally certain trade-offs with our approach to download full models to obtain personalized federated updates with regard to privacy, and subsequently added an additional experimental section (4.3) to characterize FedFomo’s performance with local differential privacy. Regarding bandwidth, we currently reference various methods orthogonal to our approach such as model distillation, compression, and network pruning (Chen et al 2017, Hinton et al 2015) that can be incorporated as complementary procedures to reduce the total download footprint. We also addressed the privacy concern by adding a new section and empirical analysis of our method training with local differential privacy (Locally Private FedFomo, Sec. 4). We report results in Table 4, and we give further evidence in Figure 5 to show that we can correctly learn about the right underlying distributions even with these noisier updates.
>
> **Better motivations for contributions**
> Our weight computation is a first-order approximation of the “optimal” weights to perform a personalized weighted average over the downloaded models, where optimal is determined by the client’s target distribution, with $w^* = \arg \min \mathcal{L}_\tau(\sum_n w_n \theta_n)$ in particular. Because at each federating round the available models $\theta_n$ are fixed, we can view this as a convex optimization problem for optimal $w$, with possible alternatives being any higher order approximation or the optimal solution given a suitable loss function. However, we were encouraged to stick with the first order form and its simplicity for several reasons:
> * Loss-agnostic: because the weighting is proportional to a normalized difference between two losses, we can apply this update with any loss. Additionally, while an exact solution may exist for certain losses, we do not have to work out alternative derivations for each of them.
> * Computational efficiency. To enable much better personalization than typical federated updates (e.g. taking a simple average over all modes or weighing by the size of its local training dataset), we compute adaptive weights that may be changed based on the current model parameters. To bound this extra overhead, it is in our best interest to reduce any additional computations, where practically speaking we avoid having to do any backpropagations and can compute the weights relatively freely. This is especially important if these weight calculations are made server-side, where iterative solutions would involve back and forth communications between clients and server in a prohibitively costly manner.
> * Intuitiveness. While the weight calculation is principled as a first order approximation, it also captures the intuitive notion that if a downloaded model performs better than our current model on our target dataset, then to optimize in a personalized manner to our target distribution we should factor in that model more.
>
> We rewrote the model exchange scheme to hopefully clarify the process, and are working on including additional experimental details in the appendix.
>
> **Ablation on design parameters** We conducted an additional ablation over different values of epsilon and numbers of models downloaded, and added these results to our revision (Fig 3, last experiment in Section 4.1). We did not find a strong correlation between performance and epsilon, but observed larger number of model downloads did increase performance across epsilon (Fig 3) in our latent non-IID setup, and in general this was agreed with in Table 2. One possible reason behind why more downloads does not always guarantee higher performance is that the loss differences in our FedFomo update may be somewhat noisy, especially during earlier federation rounds when client local models are not sufficiently trained and it may then be easier for any model to outperform another model. We think this is an interesting property to pursue in future work, and thank the reviewer for this comment.

---

> > ### Author Response · Authors · 2020-11-21
> > **Clarified motivations and details, added additional design ablations, further empirical insight into method (2/2)**
> >
> > **No explicit scheme to keep the personal models of the nodes close together** We think this is an especially interesting point, as empirically we found that even without any explicit scheme to keep model weights close to each other, we outperformed other methods that do (e.g. FedProx, Li et al 2020; pFedMe, Dinh et al 2020), and weight divergence was a core problem for non-IID FL as originally presented in Zhao et al 2018. We view this as a parsimonious feature to our method.
> >
> > Regarding why the model weights do not drift apart such that no one benefits anyone in a federated update, we think this has to do with (1) how we calculate federated updates using only compatible models in the first place--where from one client’s perspective we do not incorporate the weights of other models if they performed worse on our target objective than our current model--and (2) the assumption that all models initialize with the same starting weights. Under (2), Zhao et al 2018 characterize the root of weight divergence between two models as the difference in their local training data distributions (using EMD in their case). However because in practice our method is able to recover the underlying client groups with the same data distributions and only averages those client models together, the models that are being federated together would not exhibit weight divergence through differences in their training distributions, and furthermore under standard FL training assumptions (e.g. consistent number of local epochs between rounds and learning rate across clients), this divergence is further constrained. Additionally, once a grouping is successfully found, the federated updates only bring their weights closer together, such that in future rounds we can continue to cap the potential divergence.
> >
> > **How do weights evolve over time?** We provide additional experimental results to prove more insight into this, and include a new section (Personalized model weighting, Section 4.1) to describe this. As the actual loss deltas can be stochastic at any interation, we found that the best way to understand how model weights evolve over time was through interpreting changes in the client-to-client weights, which reflect the “affinity” of clients to models. If the models drift apart, then evaluating this will decrease these values. However empirically we show that FedFomo allows client models belonging to the same distribution to remain aligned. This suggests empirically that FedFomo can “discover” underlying clusters of the local training dataset distributions while figuring out who to federate with who. We additionally added further visualizations and discussion of this in the out-of-local distribution personalized setting (Figure 4), and regarding each client's weights over time (Figure 6, Figure 7)
> >
> > **Minor remarks** We apologize for these edits and appreciate the reviewer's attention. We edited and clarified these points in the revision:
> > 1. We should be dividing by the norm of this vector difference
> > 2. We do not encounter division by zero errors because if max(0, loss delta) = 0 for a downloaded model, we ignore the model, e.g. we only do the normalization over the positive delays.
> > 3. This should be “...where given a local objective, clients (1) evaluate how well their received models perform on their target task, and (2) use these respective performances to weight each model’s parameters...”
> > 4.  We wanted to give notation suggesting that the clients, their models, and their local datasets are not one-to-one with the distributions (e.g. 3 clients could share the same data distribution), but realized we later used $j$ to represent an alternative client / local dataset / model. For now we have corrected this by omitting any subscript, but are open to suggestions for clarifying this further.

---

> > > ### Comment · AnonReviewer2 · 2020-11-24
> > > **Thanks**
> > >
> > > Thank you for the detailed response and improvements in the manuscript. I will increase my score from 5 to 6.

---

### Official Review · AnonReviewer1 · 2020-10-29
**Interesting approach, comparison with clustered FL approaches missing, insights limited**

**Rating:** 6
**Confidence:** 4

**Review:**

The paper proposes a new FL method that computes in every communication round for each client a personalized model as starting point for the next round of federation. The paper defines the client-specific objective as some loss function of the weighted combination of all (or subset) models on a client-specific validation set. This personalized weighted combination of the models especially fits situations where not all clients have congruent objectives such as in non-IID settings. The paper evaluates the proposed FL algorithm on standard datasets for image classification by comparing to alternative FL methods.

Pros:
- The paper targets a relevant topic
- The proposed approach also works in out-of-client-distribution settings

Cons:
- The authors mention related clustered federated learning approaches (Sattler et al., 2020; Ghosh et al., 2020; Briggs et al., 2020; Mansour et al., 2020), but do not compare against any of these methods.

- The main advantage of the proposed method is that it also works in out-of-client-distribution settings (otherwise the performance is comparable, see Table 1), i.e., when the target distribution which is not the same as their local training data distribution. However, this section is very short and the evaluation not very insightful. I recommend to more carefully investigate this property of the proposed method.

- I would like to see some more theoretical analysis and insights. For instance, Sattler et al. 2020 proved that their clustered FL approach is able correctly identify the clustered (assuming that data was generated from K different distributions) if the empirical risk computed on the loss approaches the true risk. Can you give similar guarantees?

- You should also report the communication overhead of your method and compare it with related approaches. Downloading multiple models definitely increases overhead. Methods such as the clustered FL (Sattler et al. 2020) allow to compute specialised models for free, whereas you need to download multiple copies of the model at every communication round.

---

> ### Author Response · Authors · 2020-11-21
> **Added comparison to state-of-the-art clustered FL, additional insights to revision**
>
> Thank you for your constructive comments and suggestions on our paper, especially with regard to our approach working in out-of-client-distribution settings. We added the requested experiments comparing our work against clustered FL in the revision, along with additional experiments to lend further insight into our method.
>
> **Comparison to clustered FL**  We updated our results to perform the requested comparisons to Sattler et al. 2020, and in Tables 1, 2, and 5 show that our method outperforms clustering FL in highly non-IID settings. (avg +2.12% with 2, 3, and 4 non-IID latent splits across MNIST and CIFAR-10, and +8.09% with 5 and 10 non-IID splits, similar improvements in CIFAR-100). For each, we tuned the epsilon 1 parameter from [0.4, 1.2] and reported best results. Among the comparable work suggested, we stuck with Sattler et al. 2020 as state-of-the-art clustering federated learning method, which carries advantages over the alternatives mentioned by supporting non-convex risk functions (also allowing consistent comparison across all FL methods with the same CNN architecture) and the ability to adapt and split clients into different clusters throughout training, doing so by distinguishing congruent and incongruent settings. We note that FedFomo is more competitive with Clustered FL in lower IID settings, and think this is expected from the argument to only separate clients into clusters when datasets are clearly incongruent, otherwise it may be advantageous to allow every client to federate with each other. On the other hand, our method consistently outperforms Clustered Federated Learning when we split the data into more distributions, simulating more heterogeneous settings with higher EMD.
>
> **Investigating out-of-local-distribution personalization** Thank you for the suggestion to further investigate this relative strength of our method. To do so, we added additional experiments on the CIFAR-100 dataset to test if this would generalize beyond the current dataset, and show in Table 3 that we do convincingly allow this. Furthermore, we added additional discussion and new empirical results to lend further insight into why our method works. We do so via looking at the client-to-client weights over time in this setting (Fig. 4, Fig. 6), and show that regardless of the client's specific target distribution, FedFomo is also to upweight the clients with those local training distributions. Based on our weighting scheme, this suggests that FedFomo then achieves higher performance by constructed a helpful personalized combination of model downloads which is only dependent on clients' target task via their local validation data.
>
> **Reporting communication overhead** We agree that this is an important point to keep in mind, and updated our paper to discuss both communication and bandwidth overhead with downloading multiple models (Communication and bandwidth overhead, Section 3.1). We clarify that because the multiple model downloads per client are still sent as a single “packet” (e.g. in parallel), we maintain the same number of communications as other federated methods. Meanwhile, by restricting the number of models downloadable per client we can limit our bandwidth overhead to avoid worst case quadratic memory with respect to the number of active clients. In Tables 1 and 2, we show that we can still outperform alternatives with less model downloads (5 models vs 15 total), and included an additional experimental section discussing this as a design parameter (Figure 3, Exploration with epsilon and number of models downloaded M, Section 4).
>
> **More theoretical analysis and insights** Thank you for this helpful suggestion. While we agree that more theoretical analysis could be particularly insightful, in this revision (with respect to the requested additional baseline comparisons), we focused on a variety of empirical experiments to demonstrate the strength of our method. Along the way, we introduced more empirical evidence and discussion stemming from these new experiments to show why our method works beyond merely outperforming the presented alternatives, and hope that these provide further valuable insights. Related to the ability to correctly identify the clustered data, we added an additional section (Personalized model weighting, in Section 4.1) to show that across a good variety non-IID federated settings (different number of clients, latent data distributions, under differentially private training) FedFomo is able to discover these distributions empirically (Figure 2, Figure 5), strengthening the robustness of our method.
>
> Sattler, F., Muller, K.-R., &amp; Samek, W. (2020). Clustered Federated Learning: Model-Agnostic Distributed Multitask Optimization Under Privacy Constraints. IEEE Transactions on Neural Networks and Learning Systems, 1–13. https://doi.org/10.1109/tnnls.2020.3015958

---

> > ### Comment · AnonReviewer1 · 2020-11-24
> > **Improved paper after revision**
> >
> > Thank you very much for the detailed responses.
> > The paper has improved in the revision.
> > I have adapted my rating accordingly.

---

### Comment · ~Bochao_Liu1 · 2021-08-19
**Will direct transmission of the model to other clients expose privacy？**

Thank you very much for the inspiration that your work has brought to me. But when I was reading, I wondered whether passing the model of one client directly to other clients would expose private data. We can find a lot of works mentioning that the original data can be restored through model parameters or gradients, such as https://arxiv.org/abs/1906.08935v1.
The differential privacy added later cannot protect network parameters indefinitely. As the number of queries increases, the protection capability is lost.

---

> ### Author Response · Authors · 2021-10-22
> **Thanks for your interest! Response on privacy concerns**
>
> Hi Bochao, thanks for your interest in our work! Apologies for the delayed reply. Thanks as well for your valid point via privacy and information from the leakage from the shared individual models (even with DP-SGD incorporated into local training). We think this is an outstanding direction for future work: how can we enable this personalization (via evaluation) without clients downloading individual models. Two directions may be comparing aggregate models, or using some other signal between the uploaded models or gradients that can be compared server-side (assuming a secure or trusted server as in typical FL settings).

---

> > ### Comment · ~Bochao_Liu1 · 2022-03-30
> > **Thanks**
> >
> > Thanks for your responses

---

### Decision · Program_Chairs · 2021-01-07
**Final Decision**

**Decision:**

Accept (Poster)

**Comment:**

The paper proposes a personalized federated learning method, which personalizes by computing a weighted combination of neighboring compatible models. Reviewers uniformly liked the quality of writing and level of novelty, and agree on the relevance of the problem and solution. The solution was deemed creative and particularly impactful in the important case of heterogeneous data on each node, and experiments showed convincing improvements. The discussion between reviewers and authors was constructive and has lead to further improvements of the paper. Slight concerns remained on privacy with all models stored on the server, and breath of personalized FL benchmarks used, but reviewers agreed the contributions overall are still significant enough. Future work remains on the theory of the proposed model.